# Detection of epistasis between *ACTN3* and *SNAP-25* with an insight towards gymnastic aptitude identification

Łukasz Andrzej Płóciennik[1,2]*, Jan Zaucha[3], Jan Maciej Zaucha[4],
Krzysztof Łukaszuk[5], Marek Jóźwicki[6], Magdalena Płóciennik[2], Paweł Cięszczyk[1]

**1** Department of Physical Education, Academy of Physical Education and Sport in Gdansk, Gdansk, Pomorskie Voivodeship, Poland, **2** FitnessFitback, Pomorskie Voivodeship, Poland, **3** Department of Bioinformatics, Wissenschaftszentrum Weihenstephan, Technische Universität München, Freising, Germany, **4** Department of Haematology and Transplantation, Medical University of Gdansk, Gdansk, Pomorskie Voivodeship, Poland, **5** Faculty of Health Sciences with Institute of Maritime and Tropical Medicine, Medical University of Gdansk, Gdansk, Pomorskie Voivodeship, Poland, **6** Department of Architecture and Design, Academy of Fine Arts, Gdansk, Pomorskie Voivodeship, Poland

* lukaszplociennik.awf@wp.pl

**Data Availability Statement:** All relevant data are within the manuscript and its Supporting Information files.

## Abstract

In this study, we performed an analysis of the impact of performance enhancing polymorphisms (PEPs) on gymnastic aptitude while considering epistatic effects. Seven PEPs (*rs1815739*, *rs8192678*, *rs4253778*, *rs6265*, *rs5443*, *rs1076560*, *rs362584*) were considered in a case (gymnasts)–control (sedentary individuals) setting. The study sample comprised of two athletes' sets: 27 elite (aged 24.8 ± 2.1 years) and 46 sub-elite (aged 19.7 ± 2.4 years) sportsmen as well as a control group of 245 sedentary individuals (aged 22.5 ± 2.1 years). The DNA was derived from saliva and PEP alleles were determined by PCR, RT-PCR. Following Multifactor Dimensionality Reduction, logistic regression models were built. The synergistic effect for *rs1815739 x rs362584* reached 5.43%. The *rs1815739 x rs362584* epistatic regression model exhibited a good fit to the data (Chi-squared = 33.758, p ≈ 0) achieving a significant improvement in sportsmen identification over naïve guessing. The area under the receiver operating characteristic curve was 0.715 (Z-score = 38.917, p ≈ 0). In contrast, the additive *ACTN3* –*SNAP-25* logistic regression model has been verified as non-significant. **We demonstrate that a gene involved in the differentiation of muscle architecture–*ACTN3* and a gene, which plays an important role in the nervous system–*SNAP-25* interact.** From the perspective originally established by the Berlin Academy of Science in 1751, the matter of communication between the brain and muscles via nerves adopts molecular manifestations. **Further in-vitro investigations are required to explain the molecular details of the *rs1815739* –*rs362584* interaction.**

## Introduction

By 1798, Luigi Galvani discovered two phenomena: muscle stimulation by extrinsic electricity and a genuine potential difference between the nerve and the muscle. These findings lead his

**Funding:** The study was supported by National Science Centre of Poland (No. UMO-2017/27/B/NZ7/00204). The funder had no role in study design, data collection and analysis, decision to publish, or preparation of the manuscript.

**Competing interests:** The authors have declared that no competing interests exist.

successors to investigate the details of the electrical influence on nerve function in the context of muscle movement. By now, the scientific community has reached the molecular level of understanding the mechanisms involved and have already honed in on the genomic loci affecting athleticism. As a result, multiple single nucleotide polymorphisms (SNPs) have been implicated in affecting the aptitude for gymnastics. To move beyond simple SNP associations, genetic epistasis modeling may enhance the understanding of sports performance. Authors investigating genetic interactions typically rely only on genotype frequency odds ratios [1–3] or perform Genome-Wide Interaction Analyses (GWIA) employing tests visualized by pseudo-Manhattan plotting. So far, the matter of epistasis has been investigated for: (a) the Body Mass Index (BMI) [4]; (b) physical activity in mice [5]; (c) medical disorders in clinical studies [6]; in ischemic stroke susceptibility [7].

Variant interactions including synergy or redundancy have not yet been considered in the context of predicting athletic performance [8, 9]. Instead, the total genotype score (TGS) for distinguishing athletes has been calculated several times in different research projects [10, 11]. Unfortunately, TGS models do not consider interactions between polymorphisms, i.e., their synergy and redundancy [11]. The main strength of pure epistatic models is their potential for deciphering the genetic variation of predisposed athletes *ab initio*. Interestingly, ensemble-based classifiers [12], which are free of external attributes, have so far yielded better predictions than alternative approaches incorporating environmental effects into the model.

The genetic foundations of muscle performance are explored by mathematical modeling. While parametric techniques, such as logistic regression (LR) are limited in their ability to characterize the multivariate architecture of complex phenotypes, information theory provides a solution for quantifying the information gain between different statistical models of inference. The relative difference in Shannon entropy i.e. the Kullback-Leibler divergence (also known as information gain—IG) allows selecting the optimal approach for modeling the genetic effects on phenotype. Additionally, Multifactor Dimensionality Reduction (MDR), a non-parametric statistical technique enables detecting interactions between attributes of the model. In this work, we applied this method to detect epistasis in a set of candidate genes, Artistic gymnastics is one of many sport disciplines, which has not been extensively studied with regard to its genetic underpinnings. Notwithstanding the exact definition of the proportion of speed and strength to power output, gymnastics is definitely a highly polygenic anaerobic event, dependent on multiple, potentially interacting genetic variants.

The seven PEPs that were evaluated in this study include: (1) *rs1815739*, located within the *ACTN3* gene is involved in muscle contractions [13]; (2) *rs8192678*, located within the *PPARGC1A* gene is responsible for the variability in power output; the substitution of glycine for serine at position 428 was reported to hinder performance in endurance activities [14]; (3) *rs4253778*, located within the *PPARα* gene appears to be associated with the hypertrophic effect due to its effects on the cardiac and skeletal muscle substrate utilization [15]; (4) *rs6265*, located within the *BDNF-AS* gene is highly correlated with learning and the development of memory-related hippocampal neurons; (5) *rs5443*, located within the *GNB3* gene seems to be a candidate for explaining the variability in exercise phenotypes [16, 17]. Specifically, the proportion of the *TT* genotype is more pronounced in the top-level endurance athletes as compared with the sprinter group. Hence, G protein activity may affect the likelihood of becoming a highly-qualified endurance athlete [17]; (6) *rs1076560*, located within the *DRD2* gene can predispose athletes to better performance in Australian Rules Football; it allows for specific talent identification and has been linked with motor coordination and learning [18]; (7) *rs362584*, located within the *SNAP-25* gene was found to be associated with cognitive ability [19] and with the cognitive disorder [20]. Furthermore in 2015, Islamov et al. [21] have shown that *SNAP-25* is synthesized in the motor nerve endings, and affects motor neurons of the spinal cord. The aforementioned PEPs were analyzed

**Table 1. Model adjustment according to examined SNPs.**

| SNP ⟋ OR | *ACTN3* | *PPARGC1A* | *PPARα* | *BDNF-AS* | *GNB3* | *DRD2* | *SNAP-25* |
|---|---|---|---|---|---|---|---|
| OR1[a] | 1.043 | 1.225 | 0.475 | 1.139 | 0.999 | 0.749 | 0.808 |
| OR2[b] | 1.083 | 1.114 | 0.949 | 0.990 | 1.309 | 0.570 | 1.319 |
| Model | Multiplicative | Additive | Dominant | Dominant | Recessive | Multiplicative | Over-dominant |

[a] OR1 = odds ratio for heterozygote (Mm)/odds ratio for homozygote of major allele (MM)

[b] OR2 = odds ratio for homozygote of minor allele (mm)/ odds ratio for heterozygote (Mm).

with regard to epistasis in the context of gymnastics and evaluated in terms of their ability to discriminate between athletes and non-athletic individuals.

## Results

### Quality control of SNPs called

The minor allele frequency (MAF) for every candidate SNP was no less than 16.5%, which was the lowest value for the case of *rs4253788* (*PPARα*)–in the control group (Supplementary Material 1, S2 Table in S2 File). All of the seven genetic polymorphisms were in Hardy-Weinberg equilibrium (HWE; $H_0$: $\chi^2 \leq 6.635_{(0.01;\ 1)}$).

### Models adjustment according to genetic markers

All SNPs under consideration were coded according to the values of the odds ratios for heterozygote, homozygote of major allele and for homozygote of minor allele ($odd_{Mm}$, $odd_{MM}$, and $odd_{mm}$) extracted from contingency tables [22] (S3 File, p. 2). Data in Table 1 indicates the odds ratios obtained for different genetic models.

### Entropy analysis

Next, the statistical significance has been calculated for each polymorphism's ability to distinguish between the case (athletes) and control (non-athletes) groups. The strongest effect observed for any single locus was for *PPARGC1A*. Its normalized information gain (IG) reached the value of 0.0065 bits (0.65%). It was the largest univariate factor reducing entropy with a borderline significance at p = 0.07 (at $\chi^2$ = 5.317). Table 2 presents IGs and p-values of all genetic markers in the performed analysis:

### Multifactor dimensionality reduction

Next, a genetic dendrogram has been constructed, using Rajski's distance, Ward's method and Lance and Williams recursive algorithm (S3 File, pp. 3–4). As a consequence, synergistic (red

**Table 2. Information gain values of studied genetic attributes.**

| Measures | *ACTN3* | *PPARGC1A* | *PPARα* | *BDNF-AS* | *GNB3* | *DRD2* | *SNAP-25* |
|---|---|---|---|---|---|---|---|
| IG[a] [bit] | 0.0017 | 0.0065 | 0.0043 | 0.0017 | 0.0020 | 0.0023 | 0.0001 |
| $G^2$ | 0.665 | 5.317 | 1.681 | 0.665 | 0.782 | 0.899 | 0.039 |
| p-value | 0.717 | 0.070 | 0.431 | 0.717 | 0.676 | 0.638 | 0.981 |

[a] IG–information gain (S3 File, Eq 3)

[b] $G^2$–G-square statistics (S3 File, Eq 4).

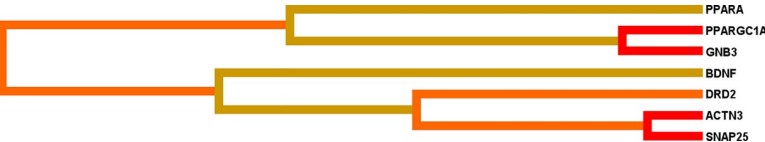

**Fig 1. A gene-gene interaction dendrogram in sports gymnastics performance[a].** [a]Orange line indicates weak positive interaction between clusters. Golden connections suggest the independence of *PPARα, BDNF*.

connections) and redundant effects have been determined ([Fig 1]). The analysis shows that polymorphisms are grouped into two clusters and two independent genetic pools of variants, namely: *PPARα, PPARGC1A –GNB3* and *BDNF, DRD2 –ACTN3 –SNAP-25*.

Epistasis between pairs of SNPs was evaluated in terms of the interaction information (I) between SNPs A and B in the context of class C: I(A; B; C), with positive values corresponding to synergy while negative values indicating a redundancy (correlation) of the markers [23]. The only strong synergistic effects were found between *ACTN3 –SNAP-25* and *PPARGC1A – GNB3*, represented by 0.0543 bits of interaction information (5.43%) and 0.0364 bits (3.64%), respectively. However, little evidence corroborates other possible two-way interactions. A positive moderation has been detected for twenty out of twenty-one combinations. The highest values regard *PPARGC1A –SNAP-25* (0.0523 bits—5.23%), *ACTN3 –PPARα* (0.298 bits—2.98%) and *GNB3 –BDNF* (0.027 bits—2.70%). The only negative interaction was between *SNAP-25* and *PPARα*; this pair of SNPs diminishes 0.0001 bits of information about sports gymnastics. The results presented above support the alternative hypothesis stipulating the existence of a synergistic effect (e.g. for *ACTN3* and *SNAP-25*) in the set comprised of twenty-one possible two-way interactions between *rs1815739, rs8192678, rs4253778, rs6265, rs5443, rs1076560, rs362584*.

Next, a filtering technique ([S3 File], Eq 8) has been applied to identify the best epistatic framework The optimal model has been obtained for the combination of *ACTN3 –PPARGC1A –PPARα–SNAP-25*. Its performance is summarized in [Table 3].

MDR analysis confirmed the statistical significance (p = 0.001) of the model by comparing the value of the sign test against 1000 random permutations of the data, assuming no association under the null hypothesis. The model achieved a balanced accuracy (weighting case and control samples so as to simulate an equal sample size in each group) of 0.712. The odds ratio of positivity within the gymnasts' group relative to the controls is equal to 6.2. Interestingly, the p-value of the model estimated from the $\chi^2$-test achieved only borderline significance, confirming previous concerns about the reliability of the p-value obtained from the MDR analysis sign-test [24]. Nevertheless, the precision is above 40% and Cohen's Kappa at 0.326 indicates a performance, which significantly surpasses naïve guessing. With regard to perfect precision and recall, the classifier is positioned in the middle of the achievable spectrum:

**Table 3. Test set results obtained for the *ACTN3 –PPARGC1A –PPARα–SNAP-25* epistatic model selected to maximize balanced accuracy in 10-fold cross validation.**

| BAL. ACC.[a] | ACC. | SENSIT. | SPECIF. | OR /CI | $\chi^2$ | $\chi^2$ p-val. | PRE.[b] | KAPPA | F[c] | CVC[d] |
|---|---|---|---|---|---|---|---|---|---|---|
| 0.712 | 0.692 | 0.75 | 0.674 | 6.211/ 0.840; 45.938 | 3.652 | = 0.056 | 0.403 | 0.326 | 0.525 | 10/ 10 |

[a]BAL. ACC.–balanced accuracy

[b] PRE.–test precision

[c] F– F1-statistics

[d]CVC–cross validation consistency (count).

F1-measure = 0.525. The training and whole data models are even more convincing (Supplementary Material 1, S5 Table, S6 Table in S2 File), since $\chi^2$ p–values retained significance after Bonferroni's correction for multiple hypothesis testing. Nevertheless, we do not have definitive evidence that that the null hypothesis can be rejected.

## Logistic regression analysis

For a simultaneous examination of the first and second order effects in the *ACTN3 – PPARGC1A –PPARα–SNAP-25* interaction, logistic regression with backward variable selection has been adopted. Since this analysis yielded empty combinations, two-way interactions were considered first. Contrasts between genotype categories were expressed in terms of cross-partial derivatives. To ensure the interpretability of the results for unbalanced classes, we used weighted effect coding (WEC). Interestingly, none of the other known mathematical and statistical coding structures apart from WEC allows detecting pure genetic interaction (Supplementary Material 1, S1 Table in S2 File). In particular, such phenomenon has been confirmed between *ACTN3* and *SNAP-25*, when setting the homogenous derived (alternative) allele category as the reference (Table 4):

The baseline OR for being a highly qualified gymnast equals 0.24, when carrying the most common genotype. Maximal log-likelihood for the estimated model totalled -133.857 with $\chi^2$-score of 34.344 (df = 8) and p-value ≈ 0.000. Although the model explains genetic foundations for sub-elite versus elite gymnasts' recognition in just 11% (pseudo $R^2$ = 0.114), we accept the global alternative hypothesis–$H1_e$, which states that at least one product term between PEPs is significantly different than zero. Considering the WEC data arrangement, the main effects of the model can be considered as non-significant being an order of magnitude less than the interaction weights, which are all below or equal $0.05^*$. Thus, individual beta weights ($b_i$) for *ACTN3* and *SNAP-25* are ≈ 0 and obeying statistical parsimony, we reject the null hypothesis. Next, we performed logistic regression for *rs1815739* and *rs362584* without first-order effects. Typically, in WEC, weights of regression coefficients do not change when the reference category is switched. The same applies to maximal log-likelihood statistics. Hence, we present different models (grouped according to reference genotype category) of interactions between genotypes in Table 5:

In agreement with previous results, all interaction effects from the model for *ACTN3 – SNAP-25*, with the derived (minor allele) genotype set as the weighted reference category are

**Table 4. The full *ACTN3 –SNAP-25* model with the derived allele reference category.**

| Constant / Genotypes | $b$ weights | CI 0.95 ± | St. errors | $\chi^2$ | p-values |
|---|---|---|---|---|---|
| Intercept | -1.445 | 0.337 | 0.171 | 8.448 | 0.004** |
| $b_{(ACTN3)}$ *heterozygous* (RX) | -0.082 | 0.313 | 0.159 | 0.518 | 0.471 |
| $b_{(ACTN3)}$ *ancestral* (RR) | 0.006 | 0.524 | 0.266 | 0.024 | 0.876 |
| $b_{(SNAP-25)}$ *heterozygous* (GA) | -0.064 | 0.388 | 0.197 | 0.326 | 0.568 |
| $b_{(SNAP-25)}$ *ancestral GG* | 0.089 | 0.372 | 0.189 | 0.473 | 0.492 |
| $b_{1(ACTN3),1(SNAP-25)}$ *heterozygous–heterozygous* | -0.805 | 0.317 | 0.161 | 4.351 | 0.037* |
| $b_{(ACTN3),(SNAP-25)}$ *heterozygous–ancestral* | 0.674 | 0.305 | 0.155 | 4.351 | 0.037* |
| $b_{(ACTN3),(SNAP-25)}$ *ancestral–heterozygous* | 1.39 | 0.74 | 0.376 | 3.694 | 0.055* |
| $b_{(ACTN3),(SNAP-25)}$ *ancestral–ancestral* | -0.876 | 0.386 | 0.196 | 4.479 | 0.034* |

$b_i$–SNP marginal effect; $b_{ii}$–2-way G-G interaction product term

** Significant at p≤ 0.01

* significant at p≤ 0.05 to second decimal place.

**Table 5. The *ACTN3* –*SNAP-25* interaction models.**

| Constant / Genotypes | *b* weights | CI 0.95 ± | St. errors | $\chi^2$ | p-values |
|---|---|---|---|---|---|
| Intercept | -1.445 | 0.337 | 0.171 | 8.448 | 0.004** |
| | | | | The model for the minor (*XX*, *AA*) allele reference category | |
| $b_{1,1}$ *heterozygous–heterozygous* | -0.805 | 0.317 | 0.161 | 4.351 | 0.037* |
| $b_{1,2}$ *heterozygous–ancestral* | 0.674 | 0.305 | 0.155 | 4.351 | 0.037* |
| $b_{2,1}$ *ancestral–heterozygous* | 1.39 | 0.74 | 0.376 | 3.694 | 0.055* |
| $b_{2,2}$ *ancestral–ancestral* | -0.876 | 0.386 | 0.196 | 4.479 | 0.034* |
| | | | | The model for the heterozygous reference category | |
| $b_{1,1}$ *derived–derived* | -1.377 | 0.854 | 0.434 | 3.171 | 0.075† |
| $b_{1,2}$ *derived–ancestral* | -0.099 | 1.323 | 0.672 | 0.147 | 0.701ns |
| $b_{2,1}$ *ancestral–derived* | 2.089 | 1.726 | 0.877 | 2.382 | 0.123‡ |
| $b_{1,2}$ *ancestral–ancestral* | -0.876 | 0.386 | 0.196 | 4.479 | 0.034* |
| | | | | The model for the ancestral (*RR*,*GG*) reference category | |
| $b_{1,1}$ *derived–derived* | -1.377 | 0.854 | 0.434 | 3.171 | 0.075† |
| $b_{1,2}$ *derived–heterozygous* | 0.809 | 0.535 | 0.272 | 3.179 | 0.085† |
| $b_{2,1}$ *heterozygous–derived* | 1.000 | 1.01 | 0.513 | 2.974 | 0.163‡ |
| $b_{2,2}$ *heterozygous–heterozygous* | -0.805 | 0.317 | 0.161 | 4.351 | 0.037* |

$b_{ii}$–2-way G-G interaction product term

** Significant at $p \leq 0.01$

* significant at $p \leq 0.05$ to second decimal place

† significant at $p < 0.1$

‡ significant at $p \leq 0.1$ to first decimal place

ns–non significant.

significant. Moreover, G-G homogenous derived genotype, ancestral-derived and heterozygous (*XX*,GA) interaction genotypes also show considerable effects, at the edge of the p-value threshold for statistical significance. Maximal log-likelihood for the interaction model for the homogenous derived allele reference category has reached the value of -134.150. The $\chi^2$ statistic was equal to 33.758 (df = 4) and pseudo $R^2$ = 0.112 giving a p-value < 0.00001. According to the model, the pure minor allele (*XX*,*AA*) genotype has the strongest negative influence. Thus, it determines the context for the other interactions. In our analysis, $b_{1,1}$, $b_{1,2}$, $b_{2,1}$, $b_{2,2}$ reached the p-value of 0.05 for the derived allele reference category (Table 5). The statistical significance was retained after applying Bonferroni's correction for multiple tests (p-value$_{\alpha/2}$ = 0.001). **In the light of this fact, three-way and multi-way interactions have not been examined**.

Particularly noteworthy is that the pure epistatic logistic regression model achieved much better performance as compared with the additive-only model. When removing all second-order derivatives, the maximal log-likelihood for the *rs1815739* + *rs362584* combination is -150.688 and becomes non-significant with a p-value of 0.409.

The results obtained from the MDR and LR analyses revealed a remarkable crosstalk between *ACTN3* –*SNAP-25* polymorphisms. Disappointingly, the $b_{heterozygous,heterozygous}$ and $b_{ancestral,ancestral}$ coefficients are attributed with negative weights; presumably, in both cases a low ratio of gymnasts to sedentary individuals (5/49 and 6/70, respectively) cause these effects (Supplementary Material 1, S4 Table in S2 File). Nevertheless, homogenous minor allele (*XX*, *AA*) genotype hosts represent the lowest chance of classification to the gymnast group: 0.059. Taking this genotype as the reference, the modeled *ACTN3* –*SNAP-25* interaction effects allow rejecting the null hypothesis of no interaction.

Based on the training set, the classification performance for the interaction model without additive terms, with the *XX–AA* allele reference category and multiplicative entries arranged according to WEC achieved the area under the ROC curve (AUC-ROC) of 0.715 (95% CI: 0.647–0.782; Z-score = 38.917, p-value $\approx$ 0.000) with a standard error (Se) of AUC-ROC = 0.034. The cut-off point was selected by maximizing the Youden index = TPF-FPF and was equal to 0.379 (Fig 2). Although the achieved classification accuracy offers good specificity and is already satisfactory to aid gymnasts' recognition, the Cohen's Kappa statistic is fair (27.2%) and F1-measure totals 0.498.

When applied to the test hold-out dataset (n = 36), our classifier has correctly classified four athletes and fifteen sedentary individuals, yielding an accuracy of 52.78%. This is unsatisfactory for the purpose of supporting decision-making in sub-elite or elite gymnasts' identification. The observed AUC-ROC (0.715) and measure of Se AUC-ROC (0.034), despite being highly significant (p-value $\approx$ 0.000) has limited potential to confer these genetic variants as predictors for athlete's discrimination in the light of the obtained Kappa statistics and F1-measure. Further studies comprising larger samples may assert the status of these variants as informative for the task of gymnasts' identification. However, our results do not allow rejecting the null hypothesis.

Worth reporting are other insights shed by the LR and WEC data organization for the *ACTN3 –PPARα*, *PPARGC1A –SNAP-25*, *PPARGC1A –GNB3*, *GNB3 –BDNF* interactions. The contingency table for *ACTN3 –PPARα* and *GNB3 –BDNF* exposed empty cell or singular representatives in genotype categories. Consequently, data were not processed any further for these models. Fortunately, the same did not apply, when *PPARGC1A –SNAP-25* and *PPARGC1A –GNB3* were considered. Both pairs of SNPs were annotated with four statistically

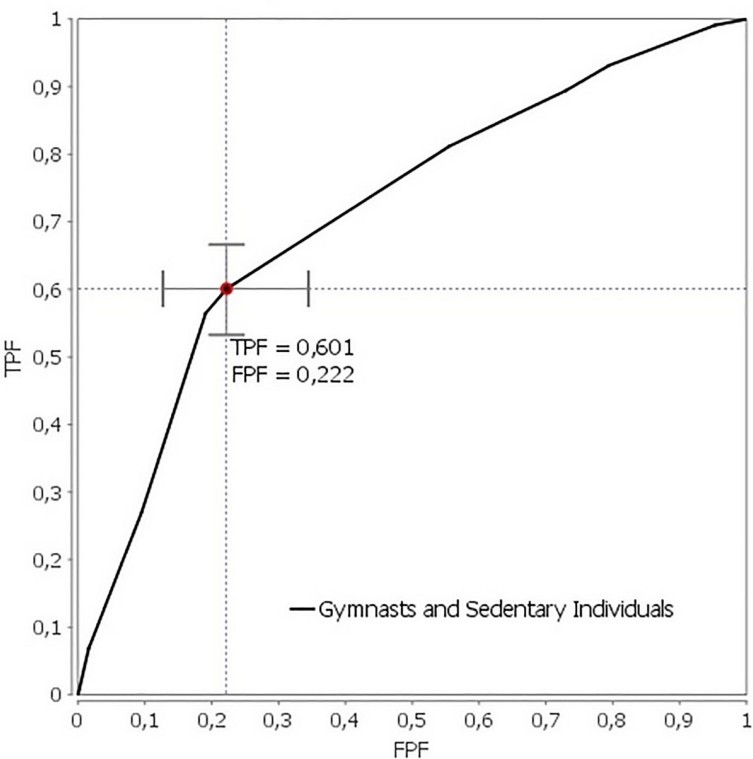

**Fig 2. The area under the curve (AUC-ROC) and cut-off point for the epistatic *rs1815739* * *rs362584* model based on the training dataset.**

significant weights (p-value ≤ 0.05) for the same second-order product terms: *PPARGC1A – SNAP-25*: $b_{GlyGly,GA}$ (*SerSer,GG* reference (ref.) genotype: favorable), *PPARGC1A –GNB3*: $b_{GlyGly,CT}$ (*SerSer,CC* ref. group: favorable), *PPARGC1A –SNAP-25*: $b_{GlyGly,GG}$ (*GlySer,GA* ref. heterozygous), *PPARGC1A –GNB3*: $b_{GlyGly,CC}$ (*GlySer,CT* heterozygous reference group), *PPARGC1A –SNAP-25*: $b_{GlySer,GG}$, $b_{SerSer,GA}$ (*GlyGly,AA* ref. disfavorable), *PPARGC1A –GNB3*: $b_{GlySer,CC}$, $b_{SerSer,CT}$ (*GlyGly,TT* reference group: disfavorable). The maximal log-likelihood value was -129.97 and -139.52, respectively. Nevertheless, the first-order effects remain insignificant for all possible pairwise combinations of SNPs. Further non-trivial effects of cross-partial G-G interactions obtained from eighteen other coding schemes applied to LR are in S2 File.

## Discussion

### The biological and sport science perspective

The ultimate goal in sport is the athletic outcome, which correlates strongly with the level of physical fitness (with psychological effects playing a secondary role). An important theoretical aspect of predicting, which individuals are genetically predisposed to athleticism regards establishing which allele encoding schemes allow for the most faithful discrimination between athletically-gifted and ungifted individuals. Apart from fundamental, molecular types of genotype ordering, we evaluated nineteen classic (statistical and mathematical) notations to describe SNPs (list available in S2 File). On the basis of planned contrasts [25], taking the trend and non-trend approaches [25], all possible ways of raw genetic data encoding have been processed to detect epistatic interactions. So far, there have been no studies in which genetic epistasis has been investigated using so many different encoding schemes. Most authors do not recognize this possibility and are reporting G-G interactions by means of LR but without considering cross-partial derivatives and using unspecified coding schemes [26, 27]. Nonetheless, a growing body of literature has discussed ways of combining non-parametric and parametric techniques with the goal of examining epistasis. A comprehensive attempt at investigating molecular interactions has been performed by Manuguerra et al. [28]. Similar to our research, these authors have presented, apart from a measure of CVC and p-values, a prediction error percentage of low and-high risk instances for given G-G models and odds ratio reports to determine the probability of false-positive predictions. Besides, it is worth noting that Wu et al. [29] have performed an analysis considering relationships between genotypes internally but also with environmental variables. Unfortunately, no information has been given on the categorical coding scheme. Only a general linear assignment was presented, which enabled us to determine the class that was used as the reference. Also, Dasgupta et al. [30], inform on gene–environmental interaction odds ratios based on MLR without considering regression coefficients. Nevertheless the essential result summarizing protective and risk-conferring alleles has been delineated. Bottema et al. applied LR to confirm interactions identified by means of MDR. Of the epistatic interactions they identified, MDR indicated that most interactions were synergistic [31]. However, the negative gene–gene interactions in the logistic regression of two-locus models suggest that polymorphisms of these genes counteract the effect of one another.

In this study we provide multiple lines of evidence indicating an interaction between *ACTN3* and *SNAP-25*. To the best of our knowledge, no previous study has reported such a relationship. Furthermore, notwithstanding the context of gymnast recognition, no data suggesting any kind of interaction between *ACTN3* and *SNAP-25* is available in String-db [32]. However, based on the outcome of the multidimensional stimulation therapy—MST intervention, neurophysiological studies have indicated the possibility of epistatic interactions between

*APOE* and *SNAP-25* [33]. Interestingly, the interaction between *ACTN3* and *APOE* has been studied to explain the potential for exceptional longevity [34]. So far, with regard to sports science, an epistasis of *ACE ID* and *ACTN3 R577X* polymorphisms has been determined, e.g. in swimmers–sprint and endurance performance [2].

In order to detect epistatic interactions Wei et al. [4] applied MLR and demonstrated two-way G-G effects affecting the body mass index (BMI) based on a genome-wide analysis. Specifically, interactions between the 19 shared epistatic genes (defined as these, which represent significant SNP interactions across cohorts) and those involving BMI candidate loci were tested across five populations (p-value < 5.0E-08). Ultimately, eight replicated SNP pairs were found in at least one cohort (p-value < 0.05) and no beta coefficients were detailed.

An interaction can also be recognized as product term, e.g. second-order parameter in logistic model under the assumption of linear coding. This technique has been used by Lee et al. [35] for testing the interaction between *EOT-2* and *CCR3* genes. The authors found that an *EOTAXIN-2* gene variant: *EOT-2+304C>A* (29L>I), was significantly associated with blood eosinophilia (p = 0.0087) by the effect of *CCR3* = -0.68. Nevertheless, no information was presented on logistic regression main effects. Potentially, an analysis of first-order parameters in the LR model may be essential to verify pseudo $R^2$ performance. In comparison all marginal weights of the full *ACTN3 –SNAP-25* model are insignificant and the benefit from applying the additive–multiplicative paradigm to gymnasts recognition is just 2‰. Likewise, the subject of interaction has been studied for the *rs12722* and the *rs13946* in *COL5A1* gene to assess a risk of the anterior cruciate ligament rupture in soccer players and controls [36]. Unfortunately, with regard to sportsman diagnosis or prognosis no details have been given on classification accuracy.

The *ACTN3 –SNAP-25* interaction allows explaining 11% of the variance between high-level sports gymnasts. Bearing in mind that genetic factors typically explain between 20% - 80% variation in a wide variety of traits relevant to athletic performance [37], the G-G epistasis detailed in this paper should not be neglected in future investigations.

## Methodological aspects

Several details of our analysis deserve particular attention. Firstly, considering the multiplicative–over-dominant scheme of epistasis between *ACTN3* and *SNAP-25*, the theoretically desirable ancestral–ancestral ($b_{ancestral,ancestral}$) or heterozygous–heterozygous ($b_{heterozygous,heterozygous}$) genotype carries a negative value. However, assuming disordinal interactions, there may be a region of non-significance [38], wherein there is a range of values for which no epistatic effect occurs. Secondly, possible signs change might occur for non-linear models even in the absence of an interaction [39]. These exist rational explanations for our results concerning $b_{heterozygous,heterozygous}$ and $b_{ancestral,ancestral}$. The third aspect concerns the data distribution. There were very few instances of gymnasts, who carried two heterozygous or dominant alleles for *ACTN3* and *SNAP-25*. An additional corroboration of our results is the fact that the *gene \* gene* interaction at the *rs1815739* and *rs362584* loci was detected by means of both: non-parametric and parametric tests. Here, after correction for multiple testing, statistical significance was far below the restrictive threshold. Finally, in terms of probability calculus, an additive only model: *ACTN3 + SNAP-25* is not significant. **Consequently, our results have interesting implications, which explain the underlying molecular details coordinating the neuromuscular system, which has been first studied by Luigi Galvani in the 18th century.** Finally, we would like to stress that further studies concerning the *ACTN3 \* SNAP-25* interactions should be conducted while considering two other levels of epistasis (suppressive, co-suppressive) [40].

### The gymnasts identification context

Despite significant results corroborating the identified genetic interaction, the resultant model for discriminating between athletes and non-athletes does not yet allow for making fully reliable predictions (Fig 2). In terms of prognosis, even a single genotype of a genetic polymorphism may be introduced as a biomarker of prevalence risk, like has been done for ischemic stroke [7]. Similarly, in our opinion, the *PPARGC1A* gene (Table 2) might be considered for diagnostic purposes. However, its usefulness in the context of gymnasts recognition has not been so far confirmed. Finally, we also observed a nominal statistical G*G partial interaction of *PPARGC1A –SNAP-25* and *PPARGC1A –GNB3* based on the gymnast status, which is interesting in the context of the studies that have associated these loci with effects relating to sport [14, 16, 17, 19–21]. Lastly, it should be acknowledged that apart from the PEPs, which we considered, interactions between other genetic loci could occur. However, expanding the analysis to include all tag SNPs (tSNPs) does not guarantee robustness for stochastic models in the aspect of predicting a predisposition to become a professional gymnast. Of note, till 2016 only twelve genetic markers have shown a positive correlation with the athlete status it at least three or more studies [41].

## Conclusions

Our analysis of seven PEPs (*ACTN3*, *PPARGC1A*, *PPARα*, *BDNF-AS*, *DRD2*, *GNB3*, *SNAP-2*), allows us to state with 93% confidence that the *rs8192678* provides as much as 0.0065 bit of information on sports gymnastics. The molecular dendrogram of gymnastics aptitude indicated the strongest connection between *rs1815739* and *rs362584*: 5.43% with a significant threshold of $\approx 0.000$, when the homogenous derived allele category is set as the reference group. According to the findings, the best MDR epistatic model of sports gymnastics comprises of: *ACTN3 –PPARGC1A –PPARα–SNAP-25* (the cross validation consistency equals 100%). Manifestly, when considering all pairwise combinations between *ACTN3*, *PPARGC1A*, *PPARα*, *BDNF-AS*, *DRD2*, *GNB3*, *SNAP*-25, the results confirm that only the second order terms of sports gymnastics epistatic models are non-zero. Lastly, out of the set of *ACTN3*, *PPARGC1A*, *PPARα*, *BDNF-AS*, *DRD2*, *GNB3* and *SNAP-25* genes, the most informative epistatic classifier–*rs1815739 x rs362584* is statistically significant in the context of sportsman recognition.

## Materials and methods

### Ethic committee

The study was approved by The Pomeranian Medical University Ethics Committee, Poland (Approval number 09/KB/IV/2011). Research procedures were run according to the World Medical Association Declaration of Helsinki. An informed consent form was completed by each participant or obtained from a parent / legal guardian (in the case of minors) in accordance with current Polish, Italian and Lithuanian law.

### Participants

A Seventy three sportsman and two hundred forty five sedentary, non-active individuals met the inclusion criteria and comprised a group for this study. They had no records of metabolic, cardiovascular diseases or musculoskeletal injuries. The subjects were non-smokers and did not take any medications. The cohort participants volunteered in Poland, Italy, Lithuania between 2012 and 2017. All participants were unrelated European men (59.4%) or women (40.6%), and all of European descent (as self-reported) for $\geq 3$ generations. Therefore, the

influence of an ethnically-induced genetic skew has been minimized and the potential population stratification issues have been controlled (Study protocol, p. 4, 5 in S1 File). The study sample included 34 females and 39 males in two homogenous athletes groups–elite (25.2 ± 2.8 years old): $n_{gymnasts\,(1,1)}$ = 18 (24.7%), who had competed at an international level (European or World Championships or Olympic Games) and sub-elite–national-level athletes (19.4 ± 3.5 years old): $n_{gymnasts\,(1,2)}$ = 55 (75.3%), who performed sports gymnastics at a national level only. Contestants were classified according to the highest-level contest they had appeared in. The gymnasts were only included if they had never been tested positive by an anti-doping agency. A control group of healthy individuals $n_{controls}$ = 245; 150 males and 95 females; 22.6 ± 2.5 years old was also selected from the Polish, Italian and Lithuanian population (college students) with no background in the sport.

Controls were matched to gymnasts in ca. 1:4 ratio; adjustment consideration has been specified in the Study protocol (S1 File).

## Methods, aims and hypotheses

In the paper, a quantitative approach to analyses has been conducted. The methods of observation and diagnostic survey were used. To gather the molecular data, PCR and RT-PCR techniques have been applied.

The goals of the research were: (a) **to measure the magnitude of informative entropy of sport PEPs in artistic gymnastics with subsequent analysis of synergistic effects or redundancy between genetic variants**; (b) **to determine marginal effects and cross-partial derivatives at the level of 2-way gene-gene interactions**; and (c) **to investigate quality measures of MDR and logistic regression epistatic models for athletes recognition.**

The aims implicate the following questions: (a) How much information will be gained on artistic gymnastics after quantifying Shannon entropy of a single genetic variant? (b) Does at least one two-attribute synergistic or redundant effect exist between sport performance enhancing polymorphisms? (c) Will the best MDR epistatic model of sports gymnastics achieve an outcome greater than 55% in cross validation consistency test? (d) For which combination of gene-gene models are the first and second order terms different than zero? (e) Are genetic classifiers statistically significant in the context of sportsman recognition? These questions concern six alternative hypotheses $H_1$:

(a) $H\,(S_{max}) < 1$; (b) $\bigvee_{IG(A;B;C)\in \boldsymbol{IG(A;B;C)}} I(A;\,B;\,C) \neq 0$; (c) $CVC_{max} > 55\%$; (d) $\bigvee_{b_i \in \boldsymbol{b_i}} b_i \neq 0$ and; (e) $\bigvee_{b_{ii} \in \boldsymbol{b_{ii}}} b_{ii} \neq 0$ when two SNPs are investigated in 2-way interaction model; (f) $AUC_i > 0{,}7$ *for* $i$ = 1,. . .,*m*; i-th Kappa statistic > 0.6,

where:

$H\,(S_{max})$ is the maximal value of Shannon entropy in the set of genetic polymorphisms $j$ = 1,. . .,*k*, *IG* is the information gain; $\boldsymbol{I(A;B;C)}$ is the vector of multiple mutual information results from all possible combinations in the analysis; $CVC_{max}$–the highest value obtained in cross-validation consistency (count) for epistatic models; $b_i$–SNP marginal effect; $b_{ii}$ is a 2-way G-G interaction product term; $AUC_i$–area under the curve for model $i$; $\bigvee_{IG(A;B;C)\in \boldsymbol{IG(A;B;C)}}$ is the existential quantifier.

## Biological sample collection and DNA extraction

The buccal cells donated by the participants were acquired using the Oragene–DNA isolation kit (DNA Genotek, Kanata, ON, Canada). The subjects abstained from drinking, and eating for 2 hours prior to saliva collection. Each participant was asked to perform a 2-min mouth rinse with water 30 min before retrieving the DNA sample. Samples were collected by passive drooling in sterile 50 ml tubes. Tubes were filled up to 4 ml, then vigorously mixed and

transported to a laboratory for further processing. All samples were stored in the same conditions at −25˚C until subsequent steps were performed.

DNA was extracted according to the producer's protocol. Briefly, the DNA material located in the Oragene tubes was incubated at 50˚C overnight. Afterward, the probes were opened and divided into four equal parts. Each one was treated with 40 μl of buffer solution supplied by the manufacturer. After a period of 10 minutes of ice incubation, centrifugation for 3 minutes at 13,000 rpm was performed. The resulting supernatant (DNA) was assessed for both purity and integrity by using spectrometric and electrophoretic methods, respectively.

## Determination of genotypes

DNA isolation and genotyping were performed in the molecular laboratory of Gdansk University of Physical Education and Sport, Poland. The genotyping error was assessed as 1%, while the call rate was above 95%. Details on PEPs genotyping can be verified in S1 File. Briefly, six gene variants (*ACTN3 –rs1815739, PPARGC1A –rs8192678, PPARα–rs4253778, BDNF-AS–rs6265, GNB3 –rs5443, DRD2– rs1076560*) were assessed by PCR. In accordance with [2], amplification was performed in a total volume of 10 μl PCR reaction mix containing 1.5 mM MgCl$_2$, 0.75 nM of each deoxynucleoside triphosphate–dNTP (Novazym, Poland), 4 pM of specific primer (Genomed, Poland) in TE (pH = 8.0; Thermo Fisher Scientific), 0.5 U DNA recombinant Taq polymerase in buffer (pH = 8.0; Sigma, Germany), 1x PCR buffer (pH = 8.7; Sigma, Germany) and 1 μl (30–50 ng) of template DNA (isolate). The thermal-time PCR amplification cycling profile conditions consisted of 10 min of preincubation at 95˚C (activation of the Taq DNA polymerase), followed by 40 cycles of denaturation at 95˚C for 15 s, and primer annealing, and extension for 1 min at 60˚C, followed by a final elongation cycle at 72˚C for 3 min. The PCR fragments were subsequently digested with the appropriate restriction enzyme. The PCR products were separated by electrophoresis at 80mV on a 2% agarose gel, stained withn DMS in DMSO ethidium bromide (250ng / ml), and visualized in UV light. The *SNAP-25* (*rs362584*) was genotyped in two replicates with TaqMan fluorescent oligonucleotide probes. Likewise, following [42], a BioRad CFX96 Touch™ RT-PCR Detection System in tandem with the Bio-Rad CFX Manager Software was used to detect the fluorescent signals and to produce a graphical representation which allowed for A / G allelic discrimination. Freshly purified / sterile water was used as a negative control for PCR.

## Statistical analyses

From 318 observations, 36 (roughly 10%) of instances were included into the test set (hold-out dataset). Minor allele frequencies were computed for each of the seven SNPs and Hardy-Weinberg equilibrium was tested. In the standard–linear approach, genotypes were coded as '1': potentially disfavorable for strength / power sports activities, '2': heterozygotes, or '3' (Supplementary Material 1, p. 8 in S2 File). Next, the most commonly used six subject-level gene models including: recessive, multiplicative, additive / harmonic, dominant, and over-dominant models [22] were computed to select the best one to the given data distribution of each SNP. After quality control of alleles and model selection, the information gain (IG) of every SNP was computed with standard coding and with the adjustment for the optimal genetic model. Next, the Multifactor Dimensionality Reduction (MDR) and logistic regression algorithms were applied.

All statistical analyses were run in MS Excel on a standard PC and in MDR program available on the Internet (https://www.multifactordimensionalityreduction.org/). The threshold for statistical significance was set to p-value ≤ 0.05, with two-sided Bonferroni correction for multiple comparisons. Formulae used for data processing have been compiled in (Theoretical background–data analysis in S3 File), for further inspection.

## Supporting information

**S1 File. Study protocol.** This protocol has been provided by the authors to give readers additional information about the research work.
(PDF)

**S2 File. Supplementary material 1.** This work contains all supplemental text, figures, and tables.
(PDF)

**S3 File. Supplementary material 2.** Theoretical background–data analysis.
(PDF)

**S4 File. Data input.**
(XLSX)

## Author Contributions

**Conceptualization:** Łukasz Andrzej Płóciennik.

**Data curation:** Łukasz Andrzej Płóciennik, Paweł Cięszczyk.

**Formal analysis:** Łukasz Andrzej Płóciennik, Magdalena Płóciennik.

**Investigation:** Łukasz Andrzej Płóciennik.

**Methodology:** Łukasz Andrzej Płóciennik, Paweł Cięszczyk.

**Project administration:** Łukasz Andrzej Płóciennik.

**Resources:** Łukasz Andrzej Płóciennik, Paweł Cięszczyk.

**Software:** Łukasz Andrzej Płóciennik.

**Supervision:** Łukasz Andrzej Płóciennik, Jan Zaucha, Jan Maciej Zaucha, Krzysztof Łukaszuk, Marek Jóźwicki.

**Validation:** Łukasz Andrzej Płóciennik.

**Visualization:** Łukasz Andrzej Płóciennik, Marek Jóźwicki.

**Writing – original draft:** Łukasz Andrzej Płóciennik, Magdalena Płóciennik.

**Writing – review & editing:** Łukasz Andrzej Płóciennik, Jan Zaucha, Jan Maciej Zaucha, Krzysztof Łukaszuk, Paweł Cięszczyk.

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
