## [Decision Letter · Decision Letter 0]

24 Jun 2020

PONE-D-20-13487

Detection of epistasis between ACTN3 and SNAP-25 with an insight towards gymnastic aptitude identification

PLOS ONE

Dear Dr. Płóciennik,

Thank you for submitting your manuscript to PLOS ONE. After careful consideration, we feel that it has merit but does not fully meet PLOS ONE’s publication criteria as it currently stands. Therefore, we invite you to submit a revised version of the manuscript that addresses the points raised during the review process.

We look forward to receiving your revised manuscript.

Kind regards,

Zezhi Li, Ph.D., M.D.

Academic Editor

PLOS ONE

Journal Requirements:

We note that one or more of the authors are employed by a commercial company: FitnessFitback.

2.1. Please provide an amended Funding Statement declaring this commercial affiliation, as well as a statement regarding the Role of Funders in your study. If the funding organization did not play a role in the study design, data collection and analysis, decision to publish, or preparation of the manuscript and only provided financial support in the form of authors' salaries and/or research materials, please review your statements relating to the author contributions, and ensure you have specifically and accurately indicated the role(s) that these authors had in your study. You can update author roles in the Author Contributions section of the online submission form.

2.2. Please also provide an updated Competing Interests Statement declaring this commercial affiliation along with any other relevant declarations relating to employment, consultancy, patents, products in development, or marketed products, etc. 

Reviewers' comments:

Reviewer's Responses to Questions

**Comments to the Author**

1. Is the manuscript technically sound, and do the data support the conclusions?

Reviewer #1: Yes

Reviewer #2: Yes

2. Has the statistical analysis been performed appropriately and rigorously? 

Reviewer #1: Yes

Reviewer #2: I Don't Know

3. Have the authors made all data underlying the findings in their manuscript fully available?

Reviewer #1: Yes

Reviewer #2: No

4. Is the manuscript presented in an intelligible fashion and written in standard English?

Reviewer #1: Yes

Reviewer #2: Yes

5. Review Comments to the Author

Reviewer #1: Comments for ‘Detection of epistasis between ACTN3 and SNAP-25 with an insight towards gymnastic aptitude identification: ACTN3 – SNAP-25 interaction in the context of athleticism’.

In this study by Łukasz, et al., the researchers investigated the impact of performance enhancing polymorphisms (PEPs) on gymnastic aptitude that were based on SNPs located on several genes, in order to discuss the communication between the nerve and muscle on the level of molecular genetics and their results demonstrate that ACTN3 (rs1815739) interacts with SNAP-25 (rs362584). This is a reliable conclusion based on different computational models, though it needs to be confirmed by further experimental data. I prefer to publish it in Plos One.

Reviewer #2: In this study, the authors performed an analysis of gene-gene interaction of seven PEPs (rs1815739, rs8192678, rs4253778, rs6265, rs5443, rs1076560, rs362584) with gymnastic aptitude in a case (gymnasts)-control (sedentary individuals) setting. The rs1815739 x rs362584 epistatic regression model exhibited a good fit to the data achieving a significant improvement in sportsmen identification over naïve guessing. We have some doubt：

1.As we know that only one allele could not represent a gene. If you want to study the interaction of these 7 genes（ ACTN3，PPARGC1A，PPARα，BDNF，AS GNB3， DRD2， SNAP-25）, you should include all tag SNPs.

2.Please give the statistical power of this study.

6. PLOS authors have the option to publish the peer review history of their article (what does this mean?). If published, this will include your full peer review and any attached files.

Reviewer #1: No

Reviewer #2: No

---

## [Author Response · Author response to Decision Letter 0]

15 Jul 2020

PONE-D-20-13487

According to decision letter, REVIEWER #1 prefers to publish the manuscript in PLoS One without suggesting any changes. REVIEWER # 1 has stated: “This is a reliable conclusion based on different computational models, though it needs to be confirmed by further experimental data”. We are pleased to have received such positive feedback and herewith, we are submitting a revised version of the manuscript, which addresses all issues raised by REVIEWER #2.

Comments to the Author

(REVIEVER #2)

QUERY:

Has the statistical analysis been performed appropriately and rigorously?

ANSWER:

 All computational aspects, including mathematical and statistical background (equations, algorithms) were detailed in Supplementary Material S3 as well as in the references. Furthermore, the Reviewer is free to recalculate our findings since the data input is fully available (Supplementary Material S4).

QUERY:

Have the authors made all data underlying the findings in their manuscript fully available?

ANSWER:

Data input has been attached as Supplementary Material S4 (.xlsx).

REVIEWER’S COMMENTS TO THE AUTHORS:

COMMENT:

As we know that only one allele could not represent a gene. If you want to study the interaction of these 7 genes (ACTN3, PPARGC1A, PPARα, BDNF, AS GNB3, DRD2, SNAP-25), you should include all tag SNPs.

ANSWER:

 Indeed, one may find and compile broad list of tag SNP (tSNP) of ACTN3, PPARGC1A, PPARα, BDNF-AS, DRD2, GNB3, SNAP-25 genes [1-7]. However, in this research, we concentrated on genetic markers that were found to be linked with the elite athlete status [8]. They are regarded as performance enhancing polymorphisms (PEPs) and are believed to play a functional role in athletic performance. It is very well true that apart from the PEPs, which we considered, interactions between other genetic loci could occur. However, expanding the analysis to include all tag SNPs (tSNPs) does not guarantee robustness for stochastic models in the aspect of predicting a predisposition to become a professional gymnast. Of note, till 2016 only twelve genetic markers have shown a positive correlation with the athlete status it at least three or more studies [9] (lines 370-375 (here and below, please compare no marks version after revision)). We wish to point out that a similar PEP-focused analysis has been performed by Tringalia et al. (2014); this analysis included only five SNPs [10].

 Furthermore, we agree that any given PEP or SNP is only a part of a specific gene. However, the interaction between SNPs represents the relationship between entire genes, or de facto their molecular products i.e. the proteins that are encoded by those genes. This line of thinking does not deviate from what is generally accepted in the scientific community. Here, we present a couple of citation records: „ … interactions between genetic variants, that is gene–gene interactions …” [11], “… interactions between genetic variants, that is, gene-gene (whether nuclear or mitochondrial) and gene-environment interactions.” [12]. In parallel, the same meaning and terminology have been applied in the main text of our study: “An additional corroboration of our results is the fact that the gene * gene interaction at the rs1815739 and rs362584 loci…” (lines: 350-351) and “This study confirms the interaction between variants in the ACTN3 and SNAP-25 loci.” (lines: 48-49). 

 What is more, it must be acknowledged that when studying complex traits, the potential number of interactions to be tested is enormous [13]. While, the actual size of current marker panels is typically smaller than 1,000,000 SNPs [14] there is a difficulty in setting genome-wide significance level [15]. So, reducing the search space to focus on a particular research question, allows hypotheses to be convincingly tested. For example, assuming two hundred and thirty-nine fitness-related genes [8], the total number of two-way interaction 

(k = 2) combinations is 56,882. Consequently, type zero hypothesis comprises the expression: ⋁_(b_ii∈b_ii)〖b_ii=0〗 (lines 434-443) for p-valueα = 0.05. Next, applying the Bonferroni correction to the threshold [16], yields p-value = 8.79 *10-7 (i.e. 0.05/56,882). In this study, we have obtained an even higher level of significance, p = 4.18 *10-7 (lines: 221-222). 

 Bearing in mind the details stated above, as well as similar research hypotheses tested in [18-20], we wish to argue that the suggestion of testing all tag SNPs should not be regarded as a requirement to answer the sort of questions we have asked in this study. 

COMMENT:

Please give the statistical power of this study.

ANSWER:

 The results of the power analysis have been provided in the text of the manuscript 

(line 245 and Figure 2). The true positive fraction (TPF) indicates the power and sensitivity of the discriminatory model [21]. After data processing, TPF totalled 60%, which is significantly better than naïve guessing. Importantly, many other statistics present a broader view of the ACTN3 – SNAP-25 model performance. Based on the training set, the classification performance for the ACTN3 * SNAP-25 model achieved the area under the ROC curve (AUC-ROC) of 0.715 (95% CI: 0.647 – 0.782; Z-score = 38.917, p-value ≈ 0.000) with a standard error (Se) of AUC-ROC = 0.034. The cut-off point was selected by maximizing the Youden index = TPF-FPF and was equal to 0.379 (Figure 2). Although the achieved classification accuracy offers good specificity and is already satisfactory to aid gymnasts’ recognition, the Cohen’s Kappa statistic is fair (27.2%) and F1-measure totals 0.498 (lines: 240-247).

References

1. Döring FE, Onur S, Geisen U, Boulay MR, Pérusse L, Rankinen T, et al. ACTN3 R577X and other polymorphisms are not associated with elite endurance athlete status in the Genathlete study. Journal of Sports Sciences. 2010; 28(12): 1355-1359. doi: 10.1080/02640414.2010.507675.

2. Kim JH, Shin HD, Park BL, Cho YM, Kim SY, Lee HK, et al. Peroxisome proliferator-activated receptor gamma coactivator 1 alpha promoter polymorphisms are associated with early-onset type 2 diabetes mellitus in the Korean population. Diabetologia. 2005; 48: 1323-1330. doi: 10.1007/s00125-005-1793-4.

3. Shin M-J, Kanaya AM, Krauss RM. Polymorphisms in the Peroxisome Proliferator Activated Receptor α Gene Are Associated with Levels of Apolipoprotein CIII andTriglyceride in African-Americans But Not Caucasians. Atherosclerosis. 2008; 198(2): 313-319. doi: 10.1016/j.atherosclerosis.2007.10.004. 

4. Gratacòs M, Soria V, Urretavizcaya M, González JR, Crespo JM, Bayés M, et al. A brain-derived neurotrophic factor (BDNF) haplotype is associated with antidepressant treatment outcome in mood disorders. The Pharmacogenomics Journal. 2008; 8: 101-112. doi: 10.1038/sj.tpj.6500460.

5. Zhang S, Zhang J. The Association of DRD2 with Insight Problem Solving. Frontiers in Psychology. 2016; 7: 1-8. doi: 10.3389/fpsyg.2016.01865.

6. Keers R, Bonvicini C, Scassellati C, Uher R, Placentiono A, Giovannini C, et al. Variation in GNB3 predicts response and adverse reactions to antidepressants. Psychopharm. 2011; 25(7): 867-874. doi: 10.1177/0269881110376683.

7. de Geus EJC, POlderman TC, van Belzen MJ, Heutink P. The SNAP-25 gene is associated with cognitive ability: evidence from a family-based study in two independent Dutch cohorts. Molecular Psychiatry. 2006; 11: 878-886. doi: 10.1038/sj.mp.4001868.

8. Contrò V, Schiera G, Abbruzzo A, Bianco A, Amato A, Sacco A, et al. An innovative way to highlight the power of each polymorphism on elite athletes phenotype expression. Eur J Transl Myol. 2018; 12(1): 87-92. doi: 10.4081/ejtm.2018.7186.

9. Ahmetov II, Egorova ES, Gabddrakhmanova LJ, Fedotovkaya ON. Genes and Athletic Performance: An Update. Med Sport Sci. 2016; 61: 41-54. doi: 10.1159/000445240.

10. Tringalia C, Brivioa I, Stucchia B, Silvestria I, Scuratib R, Michielon G, et al. Prevalence of a characteristic gene profile in highlevel rhythmic gymnasts. Journal of Sports Sciences. 2014; 32(14): 1409-1415. doi:10.1080/02640414.2014.893371.

11. Eynon N, Ruiz JR, Duarte JA, Birk RA, Lucia A. Genes and elite athletes: a roadmap for future research. J Physiol. 2011; 589(Pt 13): 3063-3070. doi: 10.1113/jphysiol.2011.207035.

12. Eynon N, Morán M, Birk R, Lucia A. The champions’ mitochondria: is it genetically determined? A review on mitochondrial DNA and elite athletic performance. Physiol Genomics. 2011; 43: 789-798. doi: 10.1152/physiolgenomics.00029.2011.

13. Kooperberg C, LeBlanc M. Increasing the power of identifying gene × gene interactions in genome-wide association studies. Genet Epidemiol. 2008; 32(3): 255-263. doi: :10.1002/gepi.20300.

14. Becker T, Herold C, Meeters C, Mattheisen M, Baur MP. Significance Levels in Genome-Wide Interaction Analysis (GWIA). Annals of Human Genetics. 2011; 75: 29-35. doi: 10.1111/j.1469-1809.2010.00610.x.

15. Ueki M, Tamiya G. Ultrahigh-dimensional variable selection method for whole-genome gene-gene interaction analysis. BMC Bionformatics. 2012; 13(72): 1-15. doi: 10.1186/1471-2105-13-72.

16. Jiao H, Zang Young, Zhang M, Zhang Yuan, Wang Y, Wang K, et al. Genome-Wide Interaction and Pathway Association Studies for Body Mass Index. Front Genet. 2019; 10: 1-10. doi: 10.3389/fgene.2019.00404.

17. Eynon N, Alves AJ, Sagiv M, Yamin C, Sagiv M, Meckel Y. Interaction between SNPs in the NRF2 gene and elite endurance performance. Physiol Genomics. 2010; 41(1): 78-81. doi:10.1152/physiolgenomics.00199.2009.

18. Grenda A, Leońska-Duniec A, Kaczmarczyk M, Ficek K, Król P, Cięszczyk P. Interaction Between ACE I/D and ACTN3 R557X Polymorphisms in Polish Competitive Swimmers. J Hum Kinet. 2014; 42: 127-36. doi:10.2478/hukin-2014-0067.

19. Lulińska-Kuklik E, Rahim M, Domańska-Senderowska D, Ficek K, Michałowska-Sawczyn M, Moska W, et al. Interactions between COL5A1 Gene and Risk of the Anterior Cruciate Ligament Rupture. J Hum Kinet. 2018; 62: 65-71. doi: 10.1515/hukin-2017-0177. 

20. Mawet D, Milli J, Wahhaj Z, Pelat D, Absil O, Delacroix C, et al. Fundamental Limitations of High Contrast Imaging Set by Small Sample Statistics. The Astrophysical Journal. 2014; 792(97): 1-11. doi: 10.1088/0004-637X/792/2/97.

---

## [Decision Letter · Decision Letter 1]

4 Aug 2020

Detection of epistasis between ACTN3 and SNAP-25 with an insight towards gymnastic aptitude identification

PONE-D-20-13487R1

Dear Dr. Płóciennik,

We’re pleased to inform you that your manuscript has been judged scientifically suitable for publication and will be formally accepted for publication once it meets all outstanding technical requirements.

Kind regards,

Zezhi Li, Ph.D., M.D.

Academic Editor

PLOS ONE

Additional Editor Comments (optional):

Reviewers' comments:

Reviewer's Responses to Questions

**Comments to the Author**

1. If the authors have adequately addressed your comments raised in a previous round of review and you feel that this manuscript is now acceptable for publication, you may indicate that here to bypass the “Comments to the Author” section, enter your conflict of interest statement in the “Confidential to Editor” section, and submit your "Accept" recommendation.

Reviewer #1: All comments have been addressed

Reviewer #2: All comments have been addressed

2. Is the manuscript technically sound, and do the data support the conclusions?

Reviewer #1: Yes

Reviewer #2: Yes

3. Has the statistical analysis been performed appropriately and rigorously? 

Reviewer #1: Yes

Reviewer #2: Yes

4. Have the authors made all data underlying the findings in their manuscript fully available?

Reviewer #1: Yes

Reviewer #2: Yes

5. Is the manuscript presented in an intelligible fashion and written in standard English?

Reviewer #1: Yes

Reviewer #2: Yes

6. Review Comments to the Author

Reviewer #1: I support this revised article published in PLOS One. The authors have addressed all the questions in a appropriate way.

Reviewer #2: The author's answer to the question feels quite reasonable. The research ethics and publication ethics was provided in the article.

7. PLOS authors have the option to publish the peer review history of their article (what does this mean?). If published, this will include your full peer review and any attached files.

Reviewer #1: No

Reviewer #2: No

---

## [Editor Report · Acceptance letter]

17 Aug 2020

PONE-D-20-13487R1 

Detection of epistasis between *ACTN3* and *SNAP-25* with an insight towards gymnastic aptitude identification 

Dear Dr. Płóciennik:

I'm pleased to inform you that your manuscript has been deemed suitable for publication in PLOS ONE. Congratulations! Your manuscript is now with our production department. 

Kind regards, 

on behalf of

Dr. Zezhi Li 

Academic Editor

PLOS ONE